# The Impact of Wearable Devices on Physical Activity for Chronic Disease Patients: Findings from the 2019 Health Information National Trends Survey

**DOI:** 10.3390/ijerph20010887

**Published:** 2023-01-03

**Authors:** Shiyuan Yu, Zhifeng Chen, Xiang Wu

**Affiliations:** School of Medicine and Health Management, Huazhong University of Science & Technology, 13 Hangkong Road, Wuhan 430030, China

**Keywords:** wearable devices, physical activity, chronic disease, self-management

## Abstract

Background: Wearable devices are shown to be an advanced tool for chronic disease management, but their impacts on physical activity remain uninvestigated. This study aims to examine the effect of wearable devices on physical activity in general people and chronic patients. Methods: Our sample was from the third cycle of the fifth iteration of the Health Information National Trends Survey (HINTS), which includes a total of 5438 residents. Genetic matching was used to evaluate the effect of wearable devices on physical activity in different populations. Results: (1) Both using wearable devices and using them with high frequency will improve physical activity for the whole population. (2) Wearable devices may have greater positive effects on physical activity for chronic patients. (3) Especially in patients with hypertension, high-frequency use of wearable devices can significantly improve the duration and frequency of physical activity. Conclusions: Wearable devices lead to more physical activity, and the benefit is more noticeable for chronic patients, particularly those with hypertension.

## 1. Introduction

Chronic diseases such as hypertension, diabetes, chronic obstructive pulmonary disease, heart disease, and mental illness have profound impacts on public health [1]. The Centers for Disease Control and Prevention (CDC) survey shows many Americans die from chronic diseases every year, which causes enormous economic losses [2]. Therefore, managing chronic conditions and reducing the burden of chronic diseases is an essential goal of American society [3].

Physical activity is a major contributing factor in managing chronic conditions. It can help manage obesity, the most critical risk factor for chronic conditions, and then reduce the health risks of blood pressure [4] and chronic diseases [5] such as type 2 diabetes and cardiovascular diseases [6]. Some rehabilitation programs for chronic obstructive pulmonary disease and depression include physical activity (moderate-intensity exercise or strength training, or both) [7,8]. Traditional interventions, such as providing written health information, telephone counselling, and even pedometers, have been shown to be effective but not feasible in routine clinical care [9] since they typically require excessive human resources, time [9], and other resources [10] and can only offer mechanical feedback [11].

Wearable devices can track health behaviors and states, and assist in a variety of methods aimed at improving physical activity, including real-time monitoring, feedback, action cues, and goal-setting [11,12]. Specifically, they can provide customized feedback from specially designed algorithms or health professionals by communicating information to mobile applications [10]. Some preliminary studies suggest wearable devices may improve physical activity through short-term real-time monitoring, goal setting, and action cues [13,14]. Goal setting may be the primary way to motivate wearable device users to improve their physical activity, while action cues remind participants of sedentary behavior in time [9,11].

Many of the above studies have discussed the impact of wearable devices on physical activity but only covered a small portion of the population interested in health. Further studies should focus on how these interventions benefit the most people in need [12]. Several studies have investigated the impact of wearable devices on physical activity in a variety of groups, including children and adolescents [13], middle-aged and elderly individuals [9,11,15,16], patients with metabolic syndrome [9,14], cancer patients [17], and stroke survivors [18]. However, few studies have explored the impact of wearable devices on physical activity in chronic diseases. Some only discuss the effect of a particular chronic disease such as metabolic syndrome, and other studies have focused primarily on the evidence to determine the feasibility and short-term benefits of wearable devices [10,14] but lack long-term evidence. We also note that some insignificant results may be due to the low sample size [18], and few observational studies obtain reliable effects of wearable devices on physical activity with appropriate effect inference methods.

In order to study the specific role of wearable devices in improving physical activity, from a practical perspective, we pay more attention to chronic patients because PA is more important to them. We investigated various impacts of using wearable devices on the whole population and chronic patients and analyzed how to benefit the neediest people. To overcome the shortcomings of the above research, we collected data from the Health Information National Trends Survey and used a genetic matching method to identify appropriate designs and measures.

## 2. Methods

### 2.1. Data

We use data from the Health Information National Trends Survey 5 (HINTS 5 Cycle 3). HINTS is a nationally representative survey conducted by the National Cancer Institute and continuously updated every few years since 2003. It collects data on the American public’s need for access to health-related information and health-related behaviors, perceptions, and knowledge. HINTS 5 Cycle 3 began in January 2019 and concluded in April 2019, consisting of 5438 resident samples.

#### 2.1.1. The Use of Wearable Devices

Two items (usage and frequency) assessed the independent variable, the use of wearable devices: “In the past 12 months, have you used an electronic wearable device to monitor or track your health or activity? For example, a Fitbit, Apple, or Garmin Vivo fit” (Yes or No); “In the past month, how often did you use a wearable device to track your health?” (Every day; Almost every day; 1–2 times per week; Less than once per week; I did not use a wearable device in the past month).

#### 2.1.2. Physical Activity

We used physical activity as the dependent variable. These items (frequency, duration of at least moderate intensity exercise, and frequency of strength training) were used to measure physical activity: “In a typical week, how many days do you do any physical activity or exercise of at least moderate intensity, such as brisk walking, bicycling at a regular pace, and swimming at a regular pace (do not include weightlifting)?” (1 day per week, 2 days per week, 3 days per week, 4 days per week, 5 days per week, 6 days per week, and 7 days per week); “In a typical week, outside of your job or work around the house, how many days do you take leisure-time physical activities specifically designed to strengthen your muscles such as lifting weights or circuit training (do not include cardio exercise such as walking, biking, or swimming)?” (1 day per week, 2 days per week, 3 days per week, 4 days per week, 5 days per week, 6 days per week and 7 days per week); “On the days that you do any physical activity or exercise of at least moderate intensity, how long do you typically do these activities?” Participants filled in the specific duration of the moderate intensity exercise in one day, specific to minutes.

#### 2.1.3. Covariates

Our results of effect inference may be highly sensitive to any imbalance in covariates highly correlated with the measures of using wearable devices. We identified the covariates that affect these two measures of using wearable devices through preliminary empirical research [19]. Demographic variables such as age, gender, education level, and income affect whether people use wearable devices. In addition, whether people have medical insurance, overall health condition, weight perception, and a willingness to change weight will affect whether people use wearable devices. We, therefore, consider these factors as covariates in genetic matching models.

Age, marital status, and the degree of enjoyment of the exercise affect the frequency of using wearable devices [20,21]. Therefore, we consider these covariates in genetic matching models with the frequency of people using wearable devices as the independent variable.

### 2.2. Analysis

The effects of wearable devices on physical activity were examined using genetic matching, an efficient algorithm for propensity score matching. In principle, the effect inference of randomly assigned variables is straightforward because the two groups are drawn from the same population by construction, and the assignment is independent of all baseline variables [22]. However, independent variable assignment is not randomized in the observational data so the samples for matching are imbalanced. This imbalance may lead to bias in the final estimation of the effect between wearable devices and physical activity. Propensity score matching has been developed to address this issue, which adjusts for the observed confounders and reduces the conditional bias in the estimand of interest [23]. This statistical method assumes there are no unobserved confounding variables [24]. This assumption implies conditional on the observed covariates, there are no differences in the distributions of unobserved confounders between different research groups. Its accuracy depends on specific sample distributions and the estimation and understanding of propensity values.

We used a genetic matching algorithm to maximize the balance of observed covariates. This method is nonparametric and does not depend on knowing or estimating the propensity score. Genetic matching is a multivariate matching approach that uses an evolutionary search algorithm, a genetic algorithm developed by Mebane and Sekhon to optimize covariate balance [24,25]. Rather than the manual process of modifying the propensity score and balance checking, genetic matching harnesses the automated search algorithm that iteratively checks the balance on observed confounders and directs the search toward those matches that optimize balance [24]. Hence, at the expense of computational time, the genetic matching search algorithm optimizes covariate balance to the extent possible, given the data. It has already been shown to improve covariate balance across a wide range of applications [23].

After matching, we calculate the average causal effect (ACE) as [22]:(1)$$\tau |\left(T=1\right)=E\left(Y_{i1}|T_{i}=1\right)-E(Y_{i0}|T_{i}=1)$$

Let $Y_{i1}$ denote the potential outcome for unit i if the unit uses wearable devices or high-frequently use wearable devices, and let $Y_{i0}$ denote the potential outcome for unit i in the reference group. The causal effect for observation i is defined by $\tau_{i}=Y_{i1}-Y_{i0}$. Causal inference is a missing data problem because $Y_{i1}$ and $Y_{i0}$ are never both observed. $T_{i}$ is an indicator of using wearable devices or high-frequently using wearable devices equal to 1 when i is in this regime and 0 otherwise [22].

If unit i uses wearable devices or high-frequently use wearable devices (T=1), let $Y_{i1}$ denote the observed outcome and $Y_{i0}$ is the potential outcome for the reference group.

## 3. Results

### 3.1. Descriptive Statistics

We identified and subdivided all 5734 samples into diabetes, hypertension, heart disease, chronic lung disease, and depression groups. Finally, 3412 chronic patients were identified, including 1149 with diabetes, 2390 with hypertension, 526 with heart disease, 631 with chronic lung disease, and 1139 with depression (Table A1 in Appendix A).

We also compare the difference in physical activity between the 2322 general population and the 3412 chronic patients (Figure 1 and Figure 2). Generally, chronic patients take less physical activity than general people.

Furthermore, we checked the use of wearable devices by people at different exercise levels. People who do not exercise use wearable devices less than people who exercise, but there is little difference between people with different exercise levels (Table 1).

### 3.2. Balance of Covariates

We compared two groups: (1) group A, which refers to those who used (or frequently used) wearable devices, and (2) group B, which refers to those who did not use (or did not frequently use) wearable devices. Covariates were highly unbalanced between the two groups. After genetic matching, covariates achieved a better balance than before (Table A2 and Table A3 in Appendix A).

### 3.3. The Impact of Wearable Device Use on Physical Activity

After matching, we obtained the effects of wearable device use on physical activity, as shown in Table 2. Using wearable devices did not significantly affect the duration of exercise at least moderate intensity on a single day for the whole population and those with chronic diseases. Notably, the impact on exercise frequency was significant for both the whole population and those with chronic diseases, which may encourage people to increase at least moderate intensity exercise approximately every two weeks (estimate = 0.460, *p* < 0.001 and estimate = 0.471, *p* < 0.001, respectively) and approximately one strength training every three weeks (estimate = 0.402, *p* < 0.001 and estimate = 0.363, *p* < 0.001, respectively).

We then further looked at the effects for different chronic disease patients and found that using wearable devices significantly improved weekly moderate intensity exercise frequency in diabetics (estimate = 0.557, *p* = 0.032) and depressed patients (estimate = 0.670, *p* = 0.005). In addition, the weekly strength training frequency in patients with hypertension increased significantly (estimate = 0.443, *p* = 0.003). For all chronic disease patients, using wearable devices had no significant effect on improving the duration of moderate intensity exercise. Notably, no significant effect of all physical activity variables was observed in heart condition and chronic lung disease patients.

### 3.4. The Impact of Frequency of Using Wearable Device on Physical Activity

The impacts of frequency of using wearable device on physical activity are shown in Table 3. Frequent use of wearable devices increased the exercise frequency for the whole population and especially those with chronic diseases. Notably, increased strength and exercise frequency were observed for hypertension patients.

For the whole population, the effect of high-frequent wearable device use on the duration of moderate intensity exercise is not significant, but the effects on moderate intensity exercise frequency (estimate = 0.645, *p* < 0.001) and strength exercise frequency (estimate = 0.307, *p* = 0.010) were significant. For the high-frequent wearable device use group, the effect of moderate intensity exercise was greater than that of strength exercise.

For all chronic disease patients, a difference between other significant effects is that high-frequent use of wearable devices had a significant effect on the duration of moderate intensity exercise (estimate = 11.349, *p* = 0.015), which can increase each moderate intensity exercise by approximately 11 min.

For each group of chronic patients, frequent use of wearable devices significantly improved weekly exercise at least moderate intensity for hypertension patients (estimate = 0.727, *p* = 0.004) and those with chronic lung disease (estimate = 1.322, *p* = 0.003). Moreover, the effects on the duration of moderate intensity exercise (estimate = 14.148, *p* = 0.039) and the frequency of weekly strength exercise (estimate = 0.629, *p* = 0.007) were significant for hypertension patients.

## 4. Discussion

### 4.1. Improving Physical Activity by Wearable Devices

Our results show using wearable devices will affect the frequency of at least moderate intensity exercise and strength exercise for the whole population. One explanation may be wearable devices are designed to encourage users’ self-monitoring and goal-setting behaviors, which results in increased physical activity [9,17,26]. These all constitute the “nudges” or reminding function, for example, wearable devices monitor the physical condition and will remind you when your sedentary behavior reaches a threshold. At the same time, the items in the HINTS summarize the types of smart wearable devices on the market, so the conclusion is not related to specific brand design. Generally, self-monitoring and goal-setting are the basic functions of wearable devices. All these self-management techniques improve physical activity by increasing people’s self-efficacy [27]. Additionally, our research indicates that high-frequent use of wearable devices can significantly improve physical activity, which is consistent with previous studies [2,12,26].

At the same time, we have also proved that for whole people or chronic patients, people who exercise (no matter how often they exercise) have similar situations of using wearable devices. People who originally have some exercise habits may indeed use wearable devices to increase their exercise. However, we may not rule out the reverse causality that people who exercise use wearable devices more than people who never exercise. That means that the evidence is not enough to prove using wearable devices can make people who do not exercise begin to exercise.

### 4.2. More Benefits for Chronic Patients by Wearable Devices

Among the investigated population, chronic patients have less physical activity than general people. This may be due to the fact that the unhealthy lifestyle itself is a factor leading to chronic disease [28], and the influence of illness and pain after suffering from chronic disease limits the willingness to exercise [29]. That means using wearable devices to improve PA in this study is more meaningful for chronic patients. The results of this study also confirm that using wearable devices has more benefits for chronic patients.

High-frequent use of wearable devices can significantly increase the duration of at least moderate exercise for chronic patients, which may be attributed to medical advice from doctors. Healthy adults do not need to consult a doctor before starting exercise [30], while clinicians may encourage patients with chronic diseases to engage in more physical activity [31] and provide them with some personalized suggestions based on wearable device data [32]. Some studies have confirmed a diagnosis of diabetes, heart disease, asthma, or depression does not have a significant effect on physical activity. Lack of advice from healthcare providers may explain the lack of changes in physical activity levels before and after the diagnosis of chronic diseases [31].

In addition, wearable devices also exhibit more meaningful and significant effects on physical activity for chronic patients than the general population [26]. Feedback from activity monitoring can successfully improve physical activity and bring beneficial results in chronic disease management [14]. High-frequency behavior can provide timely feedback on self-management performance. Since chronic patients may be more interested in the performance of self-directed methods, if timely feedback is effective, self-efficacy can be improved substantially to increase physical activity.

### 4.3. Different Effects on Different Chronic Patients

Using wearable devices had no significant effect on physical activity for patients with heart disease and chronic lung disease and had different effects on moderate intensity exercise frequency and strength training exercise frequency for patients with diabetes, hypertension, and depression. This may be because disease characteristics will also influence physical activity [33]. For example, due to heart or lung function deficiencies, physical activity will be limited for patients with heart disease and chronic lung disease. Although exercise at moderate intensity is beneficial for these patients [34], it also brings additional risks. When conditions are not under control, it is only appropriate for patients to engage in minor physical activity [34]. Therefore, the protective effect of physical activity on individuals with multiple chronic diseases is limited to those who can undertake current monthly physical activity with recommendations [35,36]. Wearable devices can only be used as auxiliary means, and it is difficult to play a decisive role in these situations. That is why we observed a significant but small effect of increasing exercise once every two or three weeks or increasing exercise by approximately 10 min, which is consistent with our results [11].

High-frequency use has no significant effect on physical activity in patients with diabetes, heart disease, or depression, but it does have a significant effect in hypertensive patients. One underlying explanation is that if high-frequency use results in ineffective outcomes (unable to correctly reflect the real data) or negative outcomes (showing poor results), self-efficacy may be reduced. When the patient’s self-monitoring has no beneficial or positive feedback, using wearable devices may lead to an increase in depression [12]. Another study has also shown that increasing self-efficacy through high-frequency monitoring is essential for hypertension self-management [37]. For example, improving blood pressure monitoring is an effective way to reduce blood pressure. We guess monitoring will increase physical activity if it improves self-efficacy but reduce people’s willingness to do physical activity if it reduces self-efficacy. Therefore, we speculate that compared with physical indicators of hypertension, chemical indicators of diabetes, heart disease, and depression may be difficult to correctly obtain in high-frequently use of wearable devices.

### 4.4. Strength and Limitations

This paper is among the first studies to evaluate the effects of wearable devices on physical activity for chronic patients. By assessing the impacts of wearable devices across different groups, this study demonstrated that wearable device use can improve physical activity, especially for chronic patients. These findings remind us how to maximize the effectiveness of wearable devices on physical activity, which may contribute to the development of targeted strategies for chronic disease self-management. For example, we may encourage hypertensive patients to use wearable devices more frequently because the use of wearable devices can indeed increase their physical activity and bring benefits to them in self-management. In contrast, we do not recommend that other chronic patients use wearable devices simply to increase physical activity. Especially for patients with heart disease and chronic lung disease, strictly following doctors’ advice to carry out physical activity is much more important than using wearable devices to increase physical activity.

The findings should be interpreted with caution because of the following limitations. First, due to the limitations of the secondary data used in our study, it is impossible to fully construct a research concept. For example, user behavior in wearable devices may lack continuous use time, physical activity lacks slight exercise, and five chronic diseases cannot fully represent a complete chronic disease. Second, while genetic matching did control for individual-level confounders, there may be unobserved covariates that were not included. We did not consider a priori which factors are unobserved confounders, which will cause bias. This is also a limitation of genetic matching in that it can only account for observed and observable covariates [38]. Third, the existing evidence is indeed insufficient to prove that the role of wearable devices in improving physical activity can enable people who never exercise to start exercising. We suggest that the following wearable device research can focus on how to achieve the goal of patients from never exercising to starting regular exercise, and how to make patients get used to exercising through digital medicine. Thus, the relationship between wearable devices and physical activity for chronic patients awaits future studies with more comprehensive data and further methods.

## 5. Conclusions

This study obtains three main findings. First, wearable device use will improve physical activity. Second, wearable devices indeed have more benefits in physical activity for people with chronic diseases than healthy people. Finally, wearable devices have a particularly obvious benefit for patients with hypertension. These findings not only highlight the benefits of wearable devices on physical activity but also encourage those with chronic diseases to maximize the efficiency of wearable devices. We suggest that wearable devices can develop some special functions or models in combination with chronic disease management. Formulate targeted physical activity strategies according to specific disease characteristics. Achieve the win-win of wearable devices market and chronic disease management.

## Figures and Tables

**Figure 1 ijerph-20-00887-f001:**
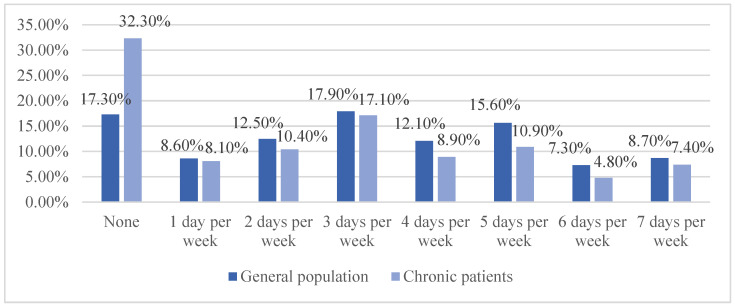
Differences in at least moderate exercise between the general population and chronic patients. The horizontal axis represents different exercise levels, and the vertical axis represents the proportion of people with each level of exercise.

**Figure 2 ijerph-20-00887-f002:**
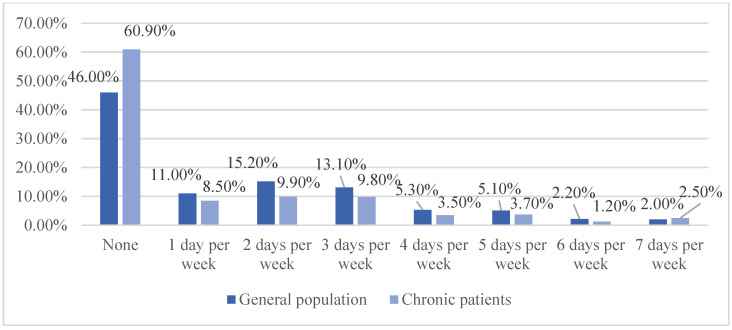
Differences in strength training between the general population and chronic patients. The horizontal axis represents different exercise levels, and the vertical axis represents the proportion of people with each level of exercise.

**Table 1 ijerph-20-00887-t001:** Proportion of people using wearable devices in each exercise group.

	Proportion of the Whole Population Using Wearable Devices in at Least Moderate Exercise	Proportion of Chronic Patients Using Wearable Devices in at Least Moderate Exercise	Proportion of the Whole Population Using Wearable Devices in Strength Training	Proportion of Chronic Patients Using Wearable Devices Strength Training
None	12.39%	11.78%	18.56%	17.10%
1 day/week	24.34%	20.69%	35.24%	31.76%
2 days/week	27.23%	26.98%	34.00%	31.52%
3 days/week	30.09%	27.08%	34.30%	28.36%
4 days/week	32.17%	27.78%	28.49%	28.72%
5 days/week	34.68%	29.93%	42.62%	40.82%
6 days/week	30.49%	29.92%	42.42%	37.93%
7 days/week	29.94%	26.80%	17.71%	14.06%

**Table 2 ijerph-20-00887-t002:** Estimation results after matching (taking wearable device use as the independent variable).

	Frequency of at Least Moderate Exercise (Times per Week)		Duration of at Least Moderate Exercise in One Day (Minute)		Frequency of Strength Training (Times per Week)	
	Estimate	*p*-Value	Estimate	*p*-Value	Estimate	*p*-Value
Whole Population	0.460	<0.001	−0.802	0.756	0.402	<0.001
Chronic Patients	0.471	<0.001	1.301	0.738	0.363	0.002
Diabetes	0.557	0.032	−1.305	0.884	0.354	0.094
Hypertension	0.328	0.058	−1.409	0.774	0.443	0.003
Heart condition	0.737	0.053	9.174	0.578	0.200	0.599
Lung disease	0.578	0.070	7.333	0.340	0.500	0.069
Depression	0.670	0.005	8.590	0.093	0.355	0.056

The *p*-values are taken from the *t*-test.

**Table 3 ijerph-20-00887-t003:** Estimation results after matching (taking the frequency of using wearable devices as the independent variable).

	Frequency of at Least Moderate Exercise (Times per Week)		Duration of at Least Moderate Exercise in One Day (Minute)		Frequency of Strength Training (Times per Week)	
	Estimate	*p*-Value	Estimate	*p*-Value	Estimate	*p*-Value
Whole Population	0.645	<0.001	5.688	0.065	0.307	0.010
Chronic Patients	0.611	<0.001	11.349	0.015	0.352	0.016
Diabetes	0.639	0.092	15.257	0.244	−0.078	0.854
Hypertension	0.727	0.004	14.148	0.039	0.629	0.007
Heart condition	1.020	0.207	24.316	0.173	0.902	0.198
Lung disease	1.322	0.003	12.280	0.320	0.354	0.412
Depression	0.426	0.213	7.497	0.202	0.269	0.330

The *p*-values are taken from the *t*-test.

## Data Availability

The data presented in this study are openly available in HINTS at https://hints.cancer.gov/ (accessed on 3 January 2022).

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
