# Peer review of "The Impact of Wearable Devices on Physical Activity for Chronic Disease Patients: Findings from the 2019 Health Information National Trends Survey"

_ijerph, 2023, doi:10.3390/ijerph20010887_

Round 1
Reviewer 1 Report
Wearable devices are shown to be an advanced tool for chronic disease management, but their impacts on physical activity remain uninvestigated.
The authors aim to examine the effect of wearable devices on physical activity in general people and chronic patients.
Their sample was from the third cycle of the fifth iteration of the Health Information National Trends Survey (HINTS), which includes a total of 5,438 residents. Genetic matching was used to evaluate the effect of wearable devices on physical activity in different populations.
The authors showed that:- both using wearable devices and using them with high frequency will improve physical activity for the whole population. -Wearable devices may have greater positive effects on physical activity for chronic patients. -Especially in patients with hypertension, high-frequency use of wearable devices can significantly improve the duration and frequency of physical activity.
The authors concluded that wearable devices lead to more physical activity, and the benefit is more noticeable for chronic patients, particularly those with hypertension
I completely am in tuning with the authors.
The manuscript is interesting.
I have some minor suggestions, with a pure academic spirit.
1. I suggest to list the results with bullet points (or number list) in the abstract.
2. Insert a better effective purpose at the end of the introduction or separated.
3. Avoid the use of the bold in the body of a section/par (see par. 2.2.1)
4. Check the MDPI standard in equation (2)
5. Check the characters in line 144-145
6. Before 2007-2008 wearable devices were specifically designed. After 2007/2008 with the introductions of Android and smartphones as we know today, the App-stores allowed the downloading of Apps in this field. I have a curiosity. How do you think this impacted on them. If you like and only if you like, I’d like a short discussion on this….
Author Response
Response to Reviewer 1 Comments (or see the attachment please)
Point 1: I suggest to list the results with bullet points (or number list) in the abstract.
Response 1: Thank you for your valuable comment. We followed your advice and have listed the results with 1) 2) 3) in the abstract (see page xx line xx) .
Point 2: Insert a better effective purpose at the end of the introduction or separated.
Response 2: Thank you for your valuable comment. Indeed, our purpose description is not clear. We have taken your suggestion and made corresponding improvements in the last paragraph of introduction (see page 2, line 62-64). The detials are as follows.
“In order to study the specific role of wearable devices in improving physical ac-tivity. And from a practical perspective, we pay more attention to chronic patients be-cause PA is more important to them.”
Point 3: Avoid the use of the bold in the body of a section/par (see part. 2.2.1)
Response 3: Thank you for pointing this out. We have fixed it and replaced the title 2.2.1.(see page 2, line 77) with “The use of wearable devices”, and explained it as an independent variable in the text (see page xx, line xx).
Point 4: Check the MDPI standard in equation (2)
Response 4: Thank you for your reminder. We have carefully checked the MDPI standard in equation (2). We have renumbered the equation and revised it’s format (see page 3) as follows.
(1)
Point 5: Check the characters in line 144-145
Response 5: Thank you for point out. We apologize that these format problems disturbed your review. We have revised the format of the elements and (in page 4, line 146-147).
Point 6: Before 2007-2008 wearable devices were specifically designed. After 2007/2008 with the introductions of Android and smartphones as we know today, the App-stores allowed the downloading of Apps in this field. I have a curiosity. How do you think this impacted on them. If you like and only if you like, I’d like a short discussion on this….
Response 6: Thank you for your willingness to discuss such a valuable question with me. In my opinion, wearable devices now refer not only to wearable hardware, but also to an application system combining software and hardware. The hardware is to collect data, while the app, hardware, provides a better human-computer interaction interface, which is easier to use in terms of visual display of information, reminders, etc. Many digital therapies can only be realized through hardware sensing and software background computing. This is the development trend of wearable devices, which is towards technology integration, convenience and intelligence.
In the past, wearable devices with only hardware could only monitor, while some intelligent analysis and reminders could not be handled automatically. After combining with the app, users can use network resources to independently realize the closed-loop from detection to intelligent result feedback. This behavior enables users to control their own sports situation in all aspects, which can improve the exercise.
Reviewer 2 Report
Quite an interesting read, I only have minor comments in the attachment. Great work.

Author Response
Response to Reviewer 2 Comments (or see the attachment please)
Point 1: There are some unidentified references or ambiguous statements in the text.
Response 1: Thank you for checking our manuscript word for word and pointing this out. We have checked our manmuscript and added more clear explanations in the marked places.
1) Add “Disease” in the title, line 3.
2) Delete “that” in page 1, line 26, page 2, line 45, page 3, line 123 and line 124, page 6, line 224 and line 226, page 7, line 255
3) Change “studies preliminarily” to “preliminary studies” in page 1, line 44.
4) “Many studies” refers to “the studies mentioned above”(in line 49).
5) The “indicates” was not correct, “disease characteristics will also influence physical activity” may be a reason of the result, and we revised ” indicates” to “because” in page 7, line 270.
6) The “ineffective” means unable to correctly reflect the real data and the “negative” means showing poor results (see page 8, line 285-286).
Point 2: What about "nudges" - for example, many products remind the wearer to stand up and meet certain PA thresholds (e.g., exercise, total calories burned).
Response 2: Thank you for your valuable comment.As you said, nudges or reminders are a kind of intuitive function. Their principles are monitoring and goal-setting. For example, the wearable device monitor your physical condition and will remind you when your sedentary behavior reaches a target value. Thank you for your reminder. We have added the description to make readers more clear about the role of nudges (see page 7, line 227-229). The description is as follows. “These all constitute the "nudges" or reminding function, for example, wearable devices monitor physical condition and will remind you when your sedentary behavior reaches threshold.”
Point 3: Anthropomorphism - studies don't show anything, authors suggest something based off of the results.
Response 3: Thank you for pointing this out. We are sorry we did not explain this clearly. The statement was expected to express that past research has shown these conclusions. The “Studies” means another referrence. Our results mention the benefits of high-frequency use. However, for some populations that do not significantly improve physical activity, it may because that high-frequency use does not bring self-efficacy to such populations. This referrence also mentioned the importance of self-efficacy in chronic disease management, so it can be regarded as the evidence of our conjecture. We have carefully considered your comment and added a more detailed explanation of our guess (see page 8, line 291-293). The guess is as follows: “We guess monitoring will increase physical activity if it improves self-efficacy but re-duce people's willing to physical activity if it reduces self-efficacy.”
Point 4: I like to use the "so what" test...meaning, you all do a great job of highlighting the findings, but "so what?" Based on the findings, what would the authors suggest people do?
Response 4: Thank you for your valuable comment. We apologize that we did not clearly clarify the significance of the study.We have carefully considered your doubt and adjusted the discussion section to more clearly explain how people can do based on our research conclusions.
The suggestions are as follows:
1) We suggest wearable devices develop some special functions for the chronic disease patients, because they are more meaningful for them to use wearable devices than the general population.
2) We suggest to formulate targeted physical activity strategies according to specific disease characteristics because different chronic disease patienets have different needs of physical activity.
And the revised content is presented below (see page 9, line 331-335): “We suggest that wearable devices can develop some special functions or models in combination with chronic disease management. Formulate targeted physical activity strategies according to specific disease characteristics. Achieve the win-win of wearable devices market and chronic disease management.”
Reviewer 3 Report
This paper claims that Chronic Patients can exercise continuously by using wearable devices. There are some unclear points in this paper, so please answer them.
1. There is a lack of discussion on the differences in exercise habits between the general population and chronic patients. Many wearable device applications devised biofeedback methods to help most people maintain exercise. I'm sure it's very beneficial for chronic patients. On the other hand, I get the impression that this paper deliberately narrowed down the scope of the analysis of the generally known law to chronic patients.
2. Data and analyzes are insufficient to reach the conclusions of this paper. The effect of using wearable devices is mentioned as the reason why moderate exercise was sustained. However, it may indicate that those who tried to exercise in chronic patients purchased wearable devices.(If the person doesn't exercise, they won't buy the wearable device.) In order to reach the conclusion of this paper, I think it is necessary to analyze whether or not people who stopped exercising were wearing wearable devices.
Author Response
Response to Reviewer 3 Comments (or see the attachment please)
Point 1: There is a lack of discussion on the differences in exercise habits between the general population and chronic patients. Many wearable device applications devised biofeedback methods to help most people maintain exercise. I'm sure it's very beneficial for chronic patients. On the other hand, I get the impression that this paper deliberately narrowed down the scope of the analysis of the generally known law to chronic patients.
Response 1: Thank you for your valuable advice. We apologize that the statement in this paper was not very clear and make you confused. We have carefully considered your comment and wanted to respond as follows. We may not deliberately narrow down the scope of the analysis. The whole population is generally divided into chronic disease population and general population. Pratically, we pay more attention to people with chronic diseases, because physical activity is more important to them. Therefore, we focused on the chronic disease population from the perspective of practical significance. Based on your valuable suggestions, we supplemented the results (Figure1 and Figure2, shown in part.3.1., in page 4, line 155-157) and the discussion (added to part.4.2, in page 7, line 244-249) on the differences of exercise habits between the general population and patients with chronic diseases.
The result is “We also compare the difference in physical activity between the 2322 general population and the 3412 chronic patients (Figure 1 and Figure 2). Generally, the chronic patients take less physical activity than the general people.”
The discussion is “Among the investigated population, chronic patients have less physical activity than general people. This may be due to the fact that the unhealthy lifestyle itself is a factor leading to chronic disease [28], and the influence of illness and pain after suf-fering from chronic disease limits the willingness to exercise [29]. That means using wearable devices to improve PA in this study are more meaningful for chronic patients. The results of this study also confirm that using wearable devices has more benefit for chronic patients.”
Point 2: Data and analyzes are insufficient to reach the conclusions of this paper. The effect of using wearable devices is mentioned as the reason why moderate exercise was sustained. However, it may indicate that those who tried to exercise in chronic patients purchased wearable devices.(If the person doesn't exercise, they won't buy the wearable device.) In order to reach the conclusion of this paper, I think it is necessary to analyze whether or not people who stopped exercising were wearing wearable devices.
Response 2: Thank you for pointing this out. We are sorry for neglecting the analysis of reverse causality. We have considered your suggenstion and would like to respond as follows. We analyze whether or not people in different exercise levels (from not exercise to exercise 7 days per week) were wearing wearable devices in Table 1. It shows that “people who exercise (no matter how often they exercise) have similar situation of using wearable devices ”(see page 4, line 158-160). The context revised is that “Besides, we checked the use of wearable devices by people at different exercise levels. People who do not exercise use wearable devices less than people who exercise, but there is little difference between people with different exercise levels (Table 1).”
We discuss it in part.4.1. (see page 7, line 236-242) that people who originally have some exercise habits may indeed use wearable devices to increase their exercise. But those who exercise in chronic patients (or general people) use wearable devices more than those who don't exercise once. The context is as followed: “At the same time, we have also proved that for the whole people or the chronic patients, people who exercise (no matter how often they exercise) have similar situa-tion of using wearable devices. People who originally have some exercise habits may indeed use wearable devices to increase their exercise. But we may not rule out the re-verse causality that people who exercise use wearable devices more than people who never exercise. That means the evidence is not enough to prove using wearable devices can make people who do not exercise begin to exercise.”
And we state it in the limitation (see page 8, line317-322) as “Third, the existing evidence is indeed insufficient to prove that the role of wearable devices in improving physical activity can enable people who never exercise to start exercising. We suggest that the following wearable devices research can focus on how to achieve the goal of patients from never exercising to starting regular exercise, and how to make patients get used to exercise through digital medicine.”
Round 2
Reviewer 3 Report
I have confirmed that your paper is sufficiently revised.